# MARL2Grid-TR: A Multi-Agent RL Benchmark in Power Grid Operations

**Enrico Marchesini**[1*]   **Eva Boguslawski**[2,3*]   **Alessandro Leite**[4]   **Christopher Amato**[5]
**Matthieu Dussartre**[3]   **Marc Schoenauer**[2]   **Benjamin Donnot**[3†]   **Priya L. Donti**[1†]

[1]Massachusetts Institute of Technology, Cambridge (MA), USA
[2]TAU, INRIA and LISN (CNRS & Univ. Paris-Saclay), Orsay, France
[3]RTE Réseau de Transport d'Électricité, Puteaux, France
[4]INSA Rouen Normandy, Univ. of Rouen Normandy, LITIS UR 4108, Rouen, France
[5]Northeastern University, Boston (MA), USA

## Abstract

Improving power grid operations is essential for enhancing flexibility and accelerating grid decarbonization. Reinforcement learning (RL) has shown promise in this domain, most notably through the Learning to Run a Power Network (L2RPN) competition series, but prior work has primarily focused on single-agent settings, neglecting the often decentralized, multi-agent nature of grid control. We fill this gap with MARL2Grid-TR, the first multi-agent RL (MARL) benchmark for grid topology and redispatching, developed in collaboration with transmission system operators. Built on RTE France's high-fidelity simulation platform, our benchmark supports decentralized control across substations and generators, with configurable agent scopes, observability settings, expert-informed heuristics, and safety-critical constraints. The benchmark includes a suite of realistic scenarios that expose key challenges, such as coordination under partial information, long-horizon objectives, and adherence to hard physical constraints. Empirical results show that current MARL methods struggle under these real-world conditions. By providing a standardized, extensible platform, we aim to advance the development of scalable, cooperative, and safe learning algorithms for power grids.[1]

## 1 Introduction

Power grid operations are undergoing a profound transformation to meet the global demands of decarbonization. The rapid rise of variable renewable energy (VRE) sources such as wind and solar requires unprecedented levels of operational flexibility and reliability. To keep the lights on while integrating VRE at scale, system operators must increasingly rely on two families of control mechanisms: (i) topology optimization, which reconfigures grid connectivity to mitigate equipment failures; and (ii) redispatching and curtailment, which adjust generators and storage units' outputs to balance supply and demand in real time. These actions play an essential role in modern grid control. However, these actions are difficult (or functionally unfeasible in the topological case) for human operators and traditional optimization-based solvers to properly handle, especially under VRE's uncertainty, and given flexible load profiles and long operating horizons (Marot et al., 2022b).

Safe and efficient grid control thus requires solving a complex, high-dimensional decision-making problem in real time. Figure 1 clarifies the setup with a simplified four-substation grid operated by two agents and interconnected by transmission lines (edges). Generators and loads are connected to buses within substations, and the power generated at each bus, which can be redispatched or curtailed, flows through the network to meet demand—the total amount of power required by the loads. Substations typically contain multiple buses that can be reconfigured via topological modifications

---

*Equal contribution; emarche@mit.edu, eva.boguslawski@rte-france.com
†Equal advising; donti@mit.edu, benjamin.donnot@rte-france.com

[1]Code is available at https://openreview.net/forum?id=mpAMH1OyMO

to modify the power flow. Both actions are subject to many physical and operational constraints: generators have ramping constraints, transmission lines have thermal capacities, and substations have switching restrictions. Violating these constraints risks blackouts or costly economic losses.

Through the "Learning to Run a Power Network competition series" (L2RPN) (Marot et al., 2022b) and the recent RL2GRID benchmark (Marchesini et al., 2025b), reinforcement learning (RL) has emerged as a promising paradigm for tackling grid control. However, these works model the problem as a single-agent task. In contrast, real-world grids are divided across multiple operators, and even within a single operator's area, the system can be decentralized. This motivates a multi-agent RL (MARL) perspective, where multiple RL agents act on different parts of the grid (U.S. DoE, 2024).

This decentralization is key for dealing with the speed and scale required to manage large amounts of VRE and flexible loads.

We present the first MARL benchmark for power grid topology and redispatching control, namely MARL2GRID-TR. Designed in collaboration with transmission system operators (TSOs) and built

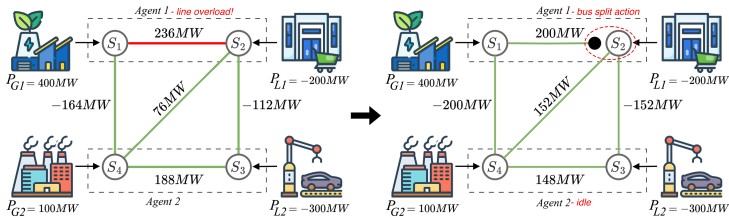

Figure 1: Toy example of a grid controlled by two agents, where all transmission lines have maximum capacities of 210 MW. A "bus split" topological (discrete) action addresses an overloaded line (red).

on the French TSO's power simulation framework (Donnot, 2020), our benchmark captures the cooperative nature of power grids. Each agent controls a subset of substations and cooperates with others to satisfy demand while maintaining grid stability. To reflect modern grid challenges, we provide multiple scenarios with different action spaces (i.e., discrete topological or continuous redispatching and curtailment actions) that scale in size and number of agents, and support various observability regimes, from fully centralized to strictly local, where agents observe only what they control. We also incorporate a multi-agent heuristic "idle" transition scheme to simplify the problem horizon under normal grid operations, and include safety-critical constraints such as load shedding, islanding, and line overloads. MARL2GRID-TR thus contributes: (i) A standardized suite of MARL tasks for discrete or continuous grid control; (ii) a PETTINGZOO interface (Terry et al., 2021), with (optional) heuristic-based transitions and constrained formalizations; and (iii) reference implementations of popular baselines for reproducible evaluation and comparison.

Overall, MARL2GRID-TR introduces a high-fidelity MARL benchmark for real power grids, providing a foundation for developing the next generation of scalable, cooperative, and safe algorithms.

## 2 PRELIMINARIES AND RELATED WORK

Table 1 shows existing environments for studying RL in energy contexts. Most prior efforts target simplified settings, such as small-scale grids, low-voltage microgrids, or building districts (Chen et al., 2022). For instance, PYTHON-MICROGRID models microgrid-level dynamics (Henri et al., 2020), GYM-ANM focuses on network management in distribution systems (Henry & Ernst, 2021), and the ARPA-E Grid Optimization competition focuses on offline optimization rather than online decision-making with RL (ARPA-E, 2023). Recently, RL2GRID (Marchesini et al., 2025b) has established a standardized RL benchmark for grid control based on the French TSO's Grid2Op, a high-fidelity power simulation framework (Donnot, 2020). Grid2Op captures crucial complexities

Table 1: Comparison of RL benchmarks for power grid operations. MARL2Grid is the only framework supporting large-scale realistic multi-agent settings for grid control with safety constraints.

| Benchmark / Environment | Scale | Multi-agent | Topology | Redisp. / Curt. | Constraints |
|---|---|---|---|---|---|
| PYTHON-MICROGRID (Henri et al., 2020) | Small | ✗ | ✗ | ✓ | ✗ |
| GYM-ANM (Henry & Ernst, 2021) | Small | ✗ | ✗ | ✓ | ✗ |
| L2RPN (Marot et al., 2020b) | Large | ✗ | ✓ | ✓ | ✗ |
| RL2GRID (Marchesini et al., 2025b) | Large | ✗ | ✓ | ✓ | ✓ |
| MARL2GRID-TR (ours) | Large | ✓ | ✓ | ✓ | ✓ |

of real grids (e.g., non-linear power flows, uncertainty from VRE, and operational constraints), and has also served as the backbone for the L2RPN competitions, which establish RL as a promising solution for grid control. However, both RL2Grid and L2RPN adopt a single-agent formulation that abstracts away the decentralized control structure of real transmission systems. Hence, they do not support varying observability regimes or coordination among multiple agents, which are essential features for scalable and practical deployment in realistic power grids. MARL2GRID-TR builds directly on the popular and realistic Grid2Op power simulation framework to address these gaps.

## 2.1 MULTI-AGENT REINFORCEMENT LEARNING

We model MARL2Grid tasks as *multi-agent Markov decision processes* (MMDPs) (Boutilier, 1996) defined by the tuple $(\mathcal{N}, \mathcal{S}, \{\mathcal{A}^i\}_{i \in \mathcal{N}}, P, R, \gamma)$, where $\mathcal{N}$ is a finite set of agents, $\mathcal{S}$ is a finite set of states, $\mathcal{A}^i$ is a set of actions for agent $i$, $P$ defines the transition dynamics over joint actions $\mathbf{a} = (a^1, \ldots, a^N)$, $R : \mathcal{S} \times \{\mathcal{A}^i\}_{i \in \mathcal{N}} \to \mathbb{R}$ is the joint reward function, and $\gamma \in [0, 1)$ is the discount factor. At each time step, each agent selects an action based on its available information, and all agents cooperate to maximize the expected discounted return $\mathbb{E}_\pi[\sum_{t=0}^{\infty} \gamma^t R(s_t, \mathbf{a}_t)]$, where $\pi = (\pi^1, \ldots, \pi^N)$ denotes the joint policy. When agents can only observe information related to the substations they control, we extend the previous definition to a *decentralized partially observable MDP* (Dec-POMDP) (Oliehoek & Amato, 2016; Åström, 1965) with $(\{\mathcal{O}^i\}_{i \in \mathcal{N}}, \{O^i\}_{i \in \mathcal{N}})$, where $\mathcal{O}^i$ is the local observation space of agent $i$ and $O^i : \mathcal{S} \times \{\mathcal{A}^i\}_{i \in \mathcal{N}} \to \Delta(\mathcal{O}^i)$ defines its observation distribution. Each agent conditions its policy $\pi^i$ on the local action-observation history $h^i$ (or, depending on the degree of partial observability, on its own observation $o_t^i \in \mathcal{O}^i$), maintaining the same objective.

**Algorithms.** A central paradigm in cooperative learning systems is *centralized training with decentralized execution* (CTDE), where agents leverage privileged information and centralized estimators during training while maintaining decentralized policies for deployment (Lowe et al., 2017). In value-based MARL, CTDE is often implemented through *value factorization*, where a centralized value function is decomposed into agent-wise utilities to guide coordination. Prominent examples include QMIX (Rashid et al., 2020) and QPLEX (Wang et al., 2021), the latter being widely adopted as a strong baseline (Papoudakis et al., 2021). CTDE has also been applied in policy-gradient methods. Algorithms such as MASAC and MAPPO (Bettini et al., 2024) employ centralized critics to stabilize learning and (potentially) improve performance, with MAPPO typically outperforming more complex approaches (Yu et al., 2022). Motivated by their widespread adoption and empirical success, MARL2GRID-TR includes QPLEX and MAPPO as representative baselines.

Power grid operations also come with safety constraints. Constrained MARL equips each agent $i \in \mathcal{N}$ with a set of auxiliary cost functions that capture constraint violations (Gu et al., 2021). Agent $i$ maintains a set of $m^i$ cost functions $\mathcal{C} := \{c_j^i\}_{j \in \{1, \ldots, m^i\}}^{i \in \mathcal{N}}$, where each $c_j^i : \mathcal{S} \times \{\mathcal{A}^i\}_{i \in \mathcal{N}} \to [0, 1]$ measures the occurrence of safety-critical events such as line overloads or load shedding. After executing $\mathbf{a}_t$ at time $t$, agents receive both task rewards and cost signals $c_j^i(s_t, \mathbf{a}_t)$. The objective is to maximize the expected return while ensuring that the cumulative discounted cost $J_j^i(\pi) := \mathbb{E}_\pi \left[ \sum_{t=0}^{\infty} \gamma^t c_j^i(s_t, \mathbf{a}_t) \right] \leq l_j^i \ \forall j \in \{1, \ldots, m^i\}$ remains below a threshold $l_j^i$ for every agent $i$ and cost index $j$. In practice, solving a constrained problem directly is difficult, causing most approaches to rely on Lagrangian relaxation. Dual variables are introduced to balance constraint satisfaction against reward maximization. Among these methods, Lagrangian MAPPO (LagrMAPPO) (Ling et al., 2022) has emerged as a strong baseline due to its simplicity, stability, and effectiveness across cooperative benchmarks (Ling et al., 2022; Aydeniz et al., 2024). For this reason, we adopt LagrMAPPO as our primary constrained baseline in MARL2GRID-TR.

## 3 MARL2GRID

In multi-agent power grid operations, each agent generally acts on a subset of substations and must coordinate with others to ensure stable long-term operation. The episodes span from one simulated week to one month, and the agents make decisions at 5-minute intervals. At each step, an agent acts on its substations and observes global or local information (based on the selected level of observability within the environment), contributing to the joint objective of maintaining safe and uninterrupted power delivery despite fluctuating demand, equipment failures, and physical constraints.

Table 2: List of base grid environments and contingencies currently supported by MARL2GRID.

| ID | Maintenance | Opponent | Subs. | Lines | Gens. | Loads | Ep. Length (steps) | \|State\| |
|---|---|---|---|---|---|---|---|---|
| **bus14** | ✓ | × | 14 | 20 | 6 | 11 | 8064 | 473 |
| **bus36** | ✓ | ✓ | 36 | 59 | 22 | 37 | 8064 | 1266 |
| **bus118** | ✓ | ✓ | 118 | 186 | 62 | 99 | 2017 | 4460 |

**Environments.** MARL2GRID-TR builds on three Grid2Op power grids (referred to as *base grids*). Table 2 summarizes their structure, and the number of substations, lines, and generators. Each base grid follows a double bus architecture, meaning that every electrical component (generators, loads, and transmission lines) can connect to one of two buses within a substation. Some environments include *Batteries (B)*, which can function both as generators (discharging) and loads (charging) in the continuous tasks. Environments also present operational contingencies designed to capture the disruptions faced by TSOs: (i) *Maintenance (M)*: Scheduled outages that agents can observe. During maintenance, a transmission line is disconnected and remains unavailable until the maintenance window ends. (ii) *Opponent (O)*: Unpredictable disturbances (e.g., weather events) causing sudden line disconnections. These events are unobserved in advance, requiring agents to react in real time. A disconnected line enters a cooldown period during which reconnection is not allowed.

Each grid in MARL2GRID-TR is partitioned among agents using the segmentation methods of Henka et al. (2022). Agents are assigned control over regions of the grid with strong internal connectivity and limited external interactions. This choice mirrors how TSOs structure control zones in practice, making our benchmark more realistic. The resulting substation-to-agent assignment for the *bus118* grid is in Table 3, while we refer to Appendix C for the remaining grid configurations. At the same time, MARL2GRID-TR is designed to be flexible. Users can modify configuration files to redefine zone assignments and explore alternative setups. Hence, the framework also supports a fully decentralized regime where every substation is controlled by its own agent. This configuration allows researchers to study the limits of coordination and scalability under higher agent counts. By supporting different configurations, MARL2GRID-TR facilitates the study of trade-offs between control and communication granularity, coordination complexity, and learning performance in multi-agent grid operations.

**Transition dynamics.** Each environment transition is driven by realistic yet synthetic time series of demand and generation, generated using ChroniX2Grid (Marot et al., 2020a).[2] At the beginning of an episode, a random timestamp is sampled to initialize the grid, ensuring exposure to varied seasonal and temporal conditions. The environment then evolves step by step in a process that mirrors real grid operations: (i) Exogenous stochastic events (e.g., weather-induced faults) are triggered according to Grid2Op's predefined probabilistic models. (ii) Agents jointly execute their topological or redispatching and curtailment actions. (iii) The system updates cooldown counters and applies any scheduled maintenance events. (iv) Grid2Op's AC power flow solver computes the new system

Table 3: Agent-to-substation assignments and dimensionality of the *bus118* grid. (T stands for the topological case, R for the redispatching and curtailment one.)

| Grid | Agent | Controlled Substations (IDs) | Lines | Gens. | Loads | \|Obs (T/R)\| | \|Actions (T/R)\| |
|---|---|---|---|---|---|---|---|
| | 0 | [0–13, 15, 116] | 23 | 7 | 12 | 281 / 187 | 414 / 5 |
| | 1 | [14, 16–18, 29, 32, 37] | 18 | 5 | 5 | 140 / 121 | 377 / 3 |
| | 2 | [33–36] | 10 | 1 | 3 | 61 / 51 | 73 / 1 |
| | 3 | [38–41, 48] | 18 | 7 | 5 | 155 / 127 | 65706 / 3 |
| | 4 | [42–47] | 10 | 1 | 6 | 84 / 146 | 52 / 2 |
| bus118 | 5 | [49–63, 65, 66] | 32 | 11 | 14 | 382 / 249 | 1375 / 13 |
| | 6 | [23, 64, 68–72] | 18 | 4 | 1 | 119 / - | 225 / - |
| | 7 | [67, 73–80, 115, 117] | 24 | 6 | 8 | 218 / 163 | 2121 / 3 |
| | 8 | [81–101] | 33 | 10 | 17 | 431 / 269 | 2640 / 10 |
| | 9 | [102–111] | 15 | 5 | 9 | 186 / 243 | 145 / 3 |
| | 10 | [19–22, 24–28, 30–31, 112–114] | 20 | 5 | 11 | 166 / 126 | 195 / 5 |
| | 11 | [0 - 117] (redispatching agent for R) | 186 | 62 | 99 | - / 1233 | - / 20 |

---

[2]We use Grid2Op's grids data, spanning up to several years and covering various conditions.

state. If the configuration is infeasible—due to islanding or unmet demand—the episode terminates. Otherwise, overloaded lines are monitored, and those exceeding limits for more than three consecutive steps are automatically disconnected. (v) Finally, all grid variables (i.e., the state) are updated, capturing the nonconvex, nonlinear, and stochastic dynamics of power systems. Depending on the observability regime, agents then receive either the full state or local observations.

**Action space.** Each base grid has two classes of tasks based on the selected action space.

For topology optimization (discrete action space), each agent can modify the topology of the substations it controls. Table 3 shows agent-substation assignments and dimensionality for the *bus118* grid. Agents can perform two types of decisions: (i) switching the status of transmission lines (i.e., connecting or disconnecting them), and (ii) reassigning electrical components to one of the two buses within a substation. While these operations correspond to simple remote switch commands in real power grids, they result in a high-dimensional space. Line switching introduces a discrete action per line, whereas bus reassignments (or "bus-splitting") yield an *exponentially* large number of valid actions. The total number of discrete actions at a double-bus substation with $N_{\text{lines}}$ lines, $N_g$ generators, and $N_l$ loads is given by (Chauhan et al., 2023): $N = 2^{N_{\text{lines}} + N_g + N_l - 1} - 1$. For example, substation #5 in Figure 2, which contains 2 generators, 1 load, and 4 lines (7 elements total), has 63 distinct topological configurations—each representing a unique combination of bus assignments. In larger grids such as *bus36*, a single substation can exceed 65,000 valid actions for a single agent. This combinatorial explosion makes traditional optimization approaches intractable and underscores the need for advanced MARL methods.

For redispatching and curtailment (continuous action space), the objective is to balance total generation and demand at every time step. To reflect real-world operations, MARL2GRID-TR introduces a mixed agent structure, where: (i) decentralized agents manage the curtailment of renewable generators and the charging/discharging of storage units within their areas, and (ii) a global redispatching agent adjusts the outputs of the other generators across the grid. The action space dimensionality thus scales linearly in the number of generators and storage units. For example, the action space size for the *bus118* grid is $N = N_{\text{redisp}} + N_{\text{curt}} + N_{\text{stor}} = 69$, where $N_{\text{redisp}}$ is the number of redispatchable generators, $N_{\text{curt}}$ the number of renewable generators, and $N_{\text{stor}}$ the number of storage units.

**State space.** The features of the state vector that are shared between the discrete and continuous tasks are listed in Table 4, including generator outputs, load demands, transmission line status and capacities.[3] In a centralized setting, each agent has access to the state (whose dimensionality is reported in Table 2). In a decentralized setting, agents observe only data corresponding to the substations they directly control. Neighboring agents share partial information for lines that connect their substations. This decentralized structure better mirrors the realities of transmission system operations, where control centers operate with limited observability and coordination. Crucially, our codebase enables users to flexibly configure observability regimes for any base grid, allowing them to extend MARL2GRID-TR and study coordination and learning under different paradigms.

**Reward function.** The objective in grid operations is to ensure long-term safety and efficiency. For topology optimization, MARL2GRID-TR adopts the reward design of Marchesini et al. (2025b), developed in consultation with TSOs. It balances three components: $R = \alpha R_{\text{survive}} + \beta R_{\text{overload}} + \eta R_{\text{cost}}$, where $\alpha$, $\beta$, and $\eta$ are weights specified in Appendix E. The three terms respectively encourage survival, penalize overloads, and account for economic costs (formal definitions are provided

Table 4: List of features composing the state of a power grid that are shared between the discrete and continuous cases. For brevity, `n_` indicates "number of", `gen` stands for "generators."

| Name(s) | Type | Dim. | Description |
|---|---|---|---|
| $\rho$ | float | `n_line` | Transmission capacity of each line |
| `gen_p` | float | `n_gen` | Gens real power |
| `load_p` | float | `n_load` | Loads active load |
| `line_status` | bool | `n_line` | Boolean flag for line connectivity |
| `timestep_overflow` | int | `n_line` | Timesteps since line exceeded capacity |

---

[3]Appendix B contains a detailed overview of the task-specific features. See RTE France (2025) for more information about these features and their ranges.

Figure 2: Overview of the multi-agent idle heuristic.

in Appendix D). For redispatching and curtailment, we adopt the reward of Donnot (2025), which directly reflects line loading margins: $R = 1 - \frac{\sum_{l \in L_c} \rho_l}{|L_c|}$, where $L_c$ is the set of connected lines and $\rho_l$ is the loading of line $l$. Specifically, grid safety decreases as line flows approach thermal limits and this formulation yields better learning performance in the continuous setting.

## 3.1 MULTI-AGENT IDLE TRANSITIONS

Given the complexity and dimensionality of the tasks, MARL2GRID-TR integrates an expert-informed *idle heuristic* (I), illustrated in Figure 2, to reduce the effective decision horizon and simplify learning. This emulation of operational behavior modifies the transition dynamics, focusing learning on safety-critical situations. Our design builds on prior L2RPN solutions and Marchesini et al. (2025b), formalizing the heuristic transitions for the multi-agent case.

For topology optimization, the heuristic issues an idle action if all line loadings $\rho$ remain below a safety threshold $\rho_{\max}$. During idle phases, agent controls are suspended and the environment progresses without intervention. When any line exceeds the threshold, control returns to the agents, who try to restore normal operation. In the redispatching and curtailment case, the heuristic first attempts to reconnect any available transmission lines. If no reconnections are possible, the heuristic performs the same idle check as in the discrete case. Importantly, the heuristic does not replace agent learning but complements it: each agent action may trigger a sequence of heuristic-guided transitions, during which rewards continue to accrue. This design combines expert-in-the-loop guidance with MARL flexibility, reducing redundant exploration, improving sample efficiency, and stabilizing training.

## 3.2 FOSTERING SAFE OPERATIONS VIA MULTI-AGENT CONSTRAINTS

MARL2GRID also includes constrained problem formalizations, in which agents have to jointly minimize safety violations under a shared set of constraints Marchesini et al. (2023). In detail, local decisions made by one agent could affect the entire grid due to the highly coupled, nonlinear, and non-convex dynamics. This phenomenon, emphasized in our discussions with TSOs at the time of development, motivated our decision to adopt a *joint constraint formulation*. Hence, constraint costs are not assigned to individual agents but are instead accumulated globally and shared among all agents—mirroring the joint reward structure. This encourages agents to reason beyond their local context and collectively maintain system-level safety, reflecting real-world operational practices. We focus on two primary classes of operational constraints, derived from major failure modes in real transmission grids, that lead to two types of constrained tasks for each *base grid*.

- *Load shedding and islanding* (L). This constraint captures two critical failure modes: (i) insufficient generation to meet demand, and (ii) the formation of electrical islands (disconnected parts of the grid). Let $P_D(s, \mathbf{a})$ and $P_G(s, \mathbf{a})$ denote the total demand and generation, respectively, given the state $s$ and the joint action $\mathbf{a}$ at a given step. We define the load shedding indicator function: $L(s, \mathbf{a}) = \mathbb{1}(P_G(s, \mathbf{a}) < P_D(s, \mathbf{a}))$, and the islanding indicator based on the number of disconnected areas $N_I(s, \mathbf{a})$ as $I(s, \mathbf{a}) = \mathbb{1}(N_I(s, \mathbf{a}) > 0)$. The per-step cost is thus defined as $C_L(s, \mathbf{a}) = L(s, \mathbf{a}) + I(s, \mathbf{a})$, and episodes are considered safe if the cumulative cost satisfies $\sum_{t=0}^{T} C_L(s, \mathbf{a}) = 0$.
- *Transmission line overload* (O). This constraint captures two key failure modes in transmission networks: (i) thermal overloads, where flows exceed line capacity, and (ii) line disconnections caused by prolonged violations. Let $P_{F,\ell}(s, \mathbf{a})$ denote the power flow on line $\ell$ at a given step, and $P_{F,\ell}^{\max}(s, \mathbf{a})$ its thermal capacity limit. We define an overload indicator function $O_\ell(s, \mathbf{a}) = \mathbb{1}(P_{F,\ell}(s, \mathbf{a}) > P_{F,\ell}^{\max}(s, \mathbf{a}))$, triggered when the line exceeds its thermal capacity, and a disconnection indicator function $D_\ell(s, \mathbf{a}) =$

$\mathbb{1}\big(\ell$ disconnected due to overload$\big)$, triggered when a line is disconnected by the environment due to sustained overload. The per-step cost across all transmission lines $\mathcal{L}$ is then $C_{\mathrm{O}}(s, \mathbf{a}) = \sum_{\ell \in \mathcal{L}} (O_\ell(s, \mathbf{a}) + D_\ell(s, \mathbf{a}))$, and the cumulative constraint is enforced as $\sum_{t=0}^{T} C_{\mathrm{O}}(s, \mathbf{a}) \leq \tau$, where $\tau$ is a fixed threshold.

By formalizing multi-agent safety constraints, we aim to provide a principled testbed for developing constrained MARL algorithms capable of balancing grid performance with operational risk.

## 4 EXPERIMENTS

We evaluate popular MARL methods that often serve as building blocks for more advanced algorithms. Consistent with prior single-agent works (Marot et al., 2022a; Marchesini et al., 2025b), topology optimization is substantially more challenging than the redispatching and curtailment setup. Due to the complexity of the task, our experiments focus primarily on the smaller *bus14* task for the topological setup, where we evaluate most algorithmic variations (e.g., the constrained algorithm) to then show how our best-performing baseline fails on the more complex *bus118* grid.[4] Specifically, we evaluate: (i) QPLEX (Wang et al., 2021), (ii) MAPPO (Yu et al., 2022) with and without the idle heuristic, and LagrMAPPO (Gu et al., 2021) (on the constrained *L* and *O* versions) in the *bus14* task; and (iii) MAPPO on the high-dimensional *bus118* task. Despite decentralization being essential to reflect how TSOs operate real grids, we also evaluate a fully observable single-agent PPO controller and its lagrangian versions LagrPPO (on the constrained *L* and *O* versions) to verify whether centralization would offer any advantage and to validate whether the challenges observed stem from the MARL decomposition or the intrinsically complex nature of the tasks. The redispatching and curtailment case is comparatively easier, and the MAPPO baseline already achieves strong performance. For this reason, we report our evaluation for this case only for the *bus118* scenario, testing MAPPO, MASAC (Bettini et al., 2024) and PPO, augmented with the idle heuristic. Crucially, these differences in the evaluation are consistent with what has been done in previous single-agent works (Marot et al., 2022a; Marchesini et al., 2025b).

Overall, this selection highlights the pressing challenges of topology optimization that motivate our benchmark, while showing that continuous redispatching, though important in practice, poses a comparatively simpler learning problem under our novel task formalization.

**Experimental setup.** Experiments were run on Xeon E5-2650 and Silver 4214R CPU nodes with 256-376GB of RAM. Baselines were implemented using custom code inspired by CleanRL's design and BenchMARL (Bettini et al., 2024), with hyperparameters selected via grid search (see Appendix E). Unless otherwise noted, results correspond to the average survival or reward of the grid over 100-episode windows, aggregated across 5 independent runs per method. Shaded regions indicate 95% bootstrapped confidence intervals. Survival is defined as the normalized fraction of time steps during which the grid remains functional, with a value of 1 indicating uninterrupted operation for a full episode. The experiments in this work required $\sim$120,000 CPU hours to execute.

### 4.1 RESULTS

**Topology Optimization (discrete).** *Overall, the baselines struggle to cope with the complexities of multi-agent topology optimization.* Figure 3 shows the training performance of the unconstrained baseline on the *bus14* grid. *MAPPO learns the most effective policy*, maintaining good operations for roughly 84% of an episode. Moreover, PPO with full observability achieves lower survival than MAPPO, showing the benefits of decentralization, and QPLEX fails to sustain stable operation beyond a few dozen steps. Augmenting these baselines with the idle heuristic converges to a $\sim$20% average survival. Hence, despite the effectiveness of the idle heuristic in multi-agent redispatching and curtailment tasks (see next section), this heuristic interacts poorly within decentralized control under a combinatorial discrete action structure. Because control is decentralized, each agent sees only a subset of the grid and must coordinate with others through the environment's nonlinear AC coupling. The idle heuristic reduces the already limited windows during which agents can experiment with (and learn) multi-step coordinated reconfigurations across zones. In an exponentially large discrete action space, where successful topological interventions are rare and require temporal coordination, this loss of actuation opportunities severely hinders exploration and joint policy

---

[4]Appendix A provides an high-level description of all the baselines.

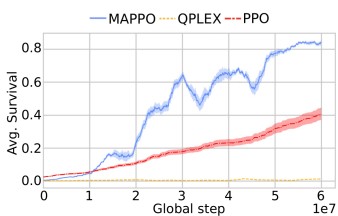

Figure 3: *bus14* (discr.): Avg. survival over training.

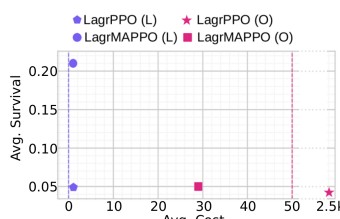

Figure 4: *bus14* (discr.): Avg. survival vs. cost at convergence for the constrained baselines.

Table 5: *bus14* (discr.): Avg. survival for the trained baselines on 2 years of test data.

| Agent type | Avg. Surv. |
|---|---|
| DoNothing | 0.18 |
| QPLEX | 0.04 |
| MAPPO | 0.79 |
| PPO | 0.38 |
| LagrMAPPO (L|O) | 0.19|0.04 |
| LagrPPO (L|O) | 0.04|0.01 |

improvement. Thus, while idle transitions accelerate learning in centralized single-agent settings (Marchesini et al., 2025b), they can become detrimental in MARL topology control due to reduced exploration capacity and the need for tightly coupled multi-agent coordination.

Figure 4 shows the Pareto frontier of average survival versus cost for LagrMAPPO and LagrPPO with both types of constraint at convergence, with dashed lines indicating the thresholds. Despite having promising constraint satisfaction results, LagrMAPPO and LagrPPO fail to achieve good performance. The best performing LagrMAPPO (L) converges to roughly $21\%$ average survival, while the single-agent baseline consistently achieves lower performance than the multi-agent counterpart. Finally, Table 5 shows the average survival at convergence for two years of data, for all baselines and for a "DoNothing" agent that only executes idle actions. These long-horizon evaluations corroborate the training curves, confirming that MAPPO achieves good control while other methods fail to maintain reliable performance.

Figure 5 analyzes how the unconstrained policies learn to control the grid in the complex discrete task (referring to Appendix F for a similar analysis for the constrained case). We report two operational metrics, *margin* and *topology*, each shown with $95\%$ confidence intervals as average scores. The margin score (defined in Section 3) measures the cumulative available capacity across all connected transmission lines. Higher values indicate that agents maintain larger safety margins and greater flexibility to handle contingencies. Successful MAPPO policies consistently maximize margins, and higher survival performance appears closely related to higher line capacity. The topology score quantifies deviations from the initial grid configuration as $-d(G_t, G_0)$, where $G_t$ is the topology at time $t$ and $d(G_t, G_0)$ is the Hamming distance from the initial configuration $G_0$. Values near 0 correspond to minimal changes, whereas increasingly negative values indicate substantial reconfigurations. Effective MAPPO agents exploit topological interventions to stabilize operation. This result is confirmed by the lower margins and topological changes of the single-agent PPO that also leads to a lower grid survival. The analysis demonstrates how these agents strike a balance between maintaining transmission margins and performing topology reconfigurations to achieve good performance.

Notably, even in the relatively small *bus14* system, the difficulty of learning safe and coordinated topological actions underscores the need for MARL advancements. This is confirmed by our additional results in Appendix F, showing how our best performing solution, MAPPO, fails at controlling the topology in the more complex *bus118* system. Notably, Marchesini et al. (2025b) shows how the single-agent

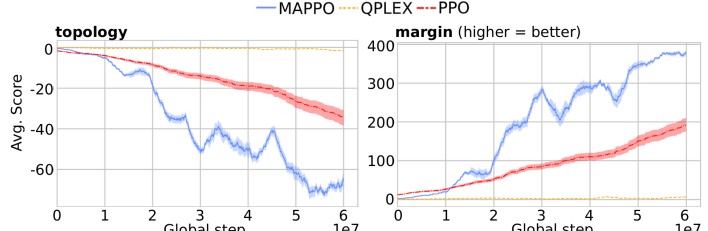

Figure 5: Avg. score for line margins (higher values mean better contingency management) and topological changes for the baselines of Figure 3.

PPO baseline (with full observability) fails on bus118, confirming that grid control challenges stem from the intrinsic structure of the topological task.

**Discrete Results Analysis.** MAPPO achieves good performance in the *bus14* setup, and our analysis of operational metrics (Figure 5) shows that good policies reliably maximize line-loading margins while performing topology reconfigurations that successfully relieve local congestion. These behaviors break down in larger grids, for which we identified four main reasons: (i) Exploration struggles in large combinatorial action spaces, where a single substation may contain tens of thousands of valid configurations, and good multi-step reconfigurations become exceedingly rare. (ii) Agents have difficulty coordinating across electrically coupled zones: actions that increase margins locally often overload distant lines (a challenge that does not appear in *bus14*). (iii) Partial observability combined with delayed, global overload penalties creates severe credit-assignment problems as agents struggle to link distant or delayed outcomes to their own actions. (iv) Topology switches involve long-horizon irreversible consequences (cooldown timers, islanding, overload-to-disconnection logic), so early random actions often lead to unrecoverable states. As a result, we notice the learned policies do not succeed in increasing margins nor in discovering meaningful topological changes in larger grids, directly explaining their poor performance. We extensively discuss avenues for future research directions related to these challenges in Section 5.

**Redispatching and curtailment (continuous).** In contrast to the topological task, *the continuous setting does not involve exponential action spaces and requires optimally balancing generation and demand, making it inherently less complex and leading to higher performance.* Figure 6 shows the learning curves of the baselines, each augmented with the heuristic from Figure 2, in the complex *bus118* grid. For this scenario, we train on February data to expose agents to more challenging operating conditions. Because the continuous task reward is defined directly in terms of margin, we report average reward rather than survival to avoid misinterpretation. Similar to the discrete case, MAPPO converges to strong performance, achieving $\sim 58\%$ average survival in our evaluation. The fully observable, single-agent PPO also achieves strong performance, but it is still inferior to MAPPO when both are trained for 3 million steps. However, in contrast to the topological setting, PPO surpasses MAPPO by 9% once trained to convergence, as shown in Table 6 (although requiring roughly 10 million steps to reach this level, underscoring its lower sample efficiency). Table 6 reports average survival over a two-year test set, comparing the baselines to the same "DoNothing" agent used in the topological case, and a "RecoPowerline" agent that directly applies the heuristic of Figure 2. Notably, MASAC is unable to achieve the performance of its heuristic, whereas MAPPO and PPO confirm their superior performance, surviving twice as long as the "DoNothing" agent.

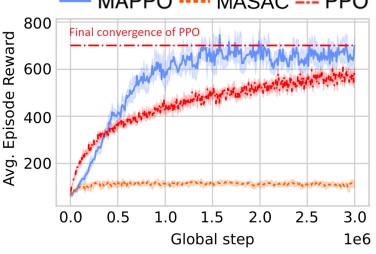

Figure 6: *bus118* (cont.) Avg. reward per episode during training.

Table 6: *bus118* (cont.) Avg. survival of the baselines on 2 years of test data.

| Agent type | Avg. Survival |
|---|---|
| DoNothing | 0.29 |
| RecoPowerline | 0.34 |
| MASAC | 0.25 |
| MAPPO | 0.58 |
| PPO | 0.67 |

## 5 CHALLENGES AND OPPORTUNITIES FOR MARL IN GRID OPERATIONS

While MARL naturally reflects the decentralized structure of real-world operations (Amato, 2024) and performs reasonably well in redispatching and curtailment tasks, our results show that popular MARL algorithms are not suitable to address high-dimensional topology optimization. This gap underscores the need for new methods and evaluation paradigms that explicitly address the combinatorial action spaces, partial observability, and safety-critical constraints of realistic and long-horizon grid operations. Closing this gap is essential if MARL is to evolve from a research prototype into a tool that supports TSOs in managing future decarbonized grids. Below, we outline key directions for such future research and how MARL can be deployed in grid operations.

**Beyond imitation.** In many domains imitation learning provides a strong starting point, but grid topology optimization lacks reliable expert demonstrations as operators themselves cannot optimally solve the problem at scale (Marot et al., 2021). This makes direct imitation infeasible. Instead, we

argue for the development of advanced heuristic-guided MARL and explainability methods (Hamman et al., 2023), where richer domain-inspired rules and approximate dynamics models serve as scaffolds to reduce exploration complexity while still allowing agents to learn effective policies.

**Coordination under partial observability.** In practice, each agent has only a local view of the grid yet must coordinate implicitly with others to prevent cascading failures. Current MARL baselines struggle to balance local autonomy with system-wide safety. Advances in communication learning, coordination graphs, and multi-agent credit assignment are needed to ensure agents act collectively rather than at cross-purposes (Marchesini et al., 2025a; Aydeniz et al., 2025).

**Scalability.** The exponential growth of topology actions poses a combinatorial barrier that is amplified in the multi-agent setting, where action spaces interact across agents. Effective abstractions (e.g., through hierarchical control, action pruning, or structured representations of topology) and exploration strategies (Marzari et al., 2025b; Marchesini & Amato, 2023) are thus crucial to scaling MARL to realistically sized grids. In MARL2Grid-TR, the 118-bus system already reaches a meaningful scale for research: it is large enough to expose the core coordination, safety, and combinatorial challenges of realistic operator-level control, yet still tractable for large-scale experimentation. Scaling to grids with thousands of buses remains an important long-term goal, but our findings indicate that substantial algorithmic advances are required before reaching that scale.

**Realism, evaluation, simulation.** Progress will also depend on more realistic evaluations. While our benchmark includes long horizons, stochastic renewable fluctuations, and safety-critical constraints, further realism is required (e.g., explicit $N-1$ security). Evaluation should also go beyond average survival to assess economic impact, robustness under rare but critical contingencies using formal tools (Liu et al., 2021; Weng et al., 2019; Marzari et al., 2025a), and cooperation in large, heterogeneous networks. Regarding simulation, the benchmark captures key operational constraints via Grid2Op's AC solver but omits fast transients, detailed inverter and protection dynamics, and some action constraints. While larger grids can be configured, MARL training on very large systems also remains computationally heavy. MARL approaches can move toward practical deployment only by coupling algorithmic advances with increasingly realistic benchmarks.

**Deployment.** While the power sector is rightly conservative, the joint development of MARL2GRID-TR with TSOs shows a clear interest in RL because traditional optimization tools struggle with the growing combinatorial and real-time complexity introduced by high VRE, frequent contingencies, and large reconfiguration spaces (Marot et al., 2020b). Crucially, RL can address these challenges and be integrated within existing operator workflows and validated through offline simulation, shadow-mode deployment, and safety filters before a broader adoption in the industry.

In summary, MARL magnifies the core challenges of grid control (e.g., combinatorial action spaces, strict safety constraints, and long horizons) while introducing new ones such as coordination under partial observability and the lack of expert demonstrations. Addressing these challenges will require going beyond standard MARL methods to design algorithms, heuristics, and evaluation protocols tailored to the unique demands of power system operations and decarbonization.

## 6 CONCLUSION

MARL2GRID-TR introduces the first multi-agent RL benchmark for realistic power grid operations, covering both discrete topology optimization and continuous redispatching, curtailment, and storage control. By distributing control across agents responsible for subsets of substations, the benchmark reflects the cooperative structure of real-world grids while exposing key challenges: partial observability, high-dimensional action spaces, and safety-critical constraints such as load shedding, islanding, and line overloads.

The benchmark provides standardized tasks of increasing complexity, PETTINGZOO-compatible interfaces, heuristic-based idle transitions, and constrained multi-agent training settings. Experiments show that while MARL achieves promising performance in a subset of the proposed tasks and is a natural paradigm for distributed grid control, current methods struggle with scalability, coordination, and safety in most of these long-horizon scenarios.

We expect MARL2GRID-TR to serve as a foundation for developing, evaluating, and comparing cooperative MARL algorithms that can enable safe and efficient grid control under modern large amounts of (distributed) VRE and flexible loads.

ACKNOWLEDGEMENTS

This work was supported in part by the AI2050 program at Schmidt Sciences (Grant G-24-66236), the MIT-IBM Watson AI Lab, the "Fondo Italiano per la Scienza" project (Grant FIS-2024-05614), and the French National Research Agency (ANR) under Grant No. ANR-23-CPJ1-0099-01. The authors thank the reviewers for their insightful and constructive feedback, which has substantially improved the quality of this work.

ETHICS STATEMENT

This work introduces a benchmark for MARL in realistic power grid operations. The benchmark is developed entirely on top of publicly available, synthetic data generated with the Grid2Op framework, ensuring that no sensitive, private, or personally identifiable information is used. The environments model stylized versions of real-world power systems in collaboration with TSOs, but do not replicate proprietary or security-critical grid infrastructure.

The primary goal of this research is to advance the development of safe, cooperative MARL methods in the context of power grid operations. While RL agents trained on our benchmark are not directly deployable in operational power grids, we acknowledge that methods for controlling critical infrastructure must be carefully validated and subject to rigorous safety and regulatory oversight before practical use. By explicitly modeling safety-critical constraints (e.g., load shedding, islanding, and line overloads), MARL2GRID-TR aims to encourage research directions that emphasize safety and reliability.

We believe that this work aligns with the ICLR Code of Ethics by supporting transparent, reproducible research and by fostering methods that can contribute positively to the reliable and decarbonized operation of power systems.

REPRODUCIBILITY STATEMENT

We have taken several steps to ensure the reproducibility of our work. The full benchmark codebase will be released as anonymous supplementary code during the review process.

Detailed descriptions of the state and action spaces, reward functions, transition dynamics, and safety constraints are provided in Section 3 and Appendices B to D, while hyperparameter choices and grid search ranges are reported in Appendix E. All experiments were run on standard CPU clusters, with hardware details and data collection protocols documented in Section 4. For each baseline, we provide references to the original algorithm and describe how it was adapted to the multi-agent power grid setting (Appendix A). Together, these materials ensure that all results presented in the paper can be independently verified and extended.

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

## A  MARL BASELINES

In this section, we briefly introduce the baseline MARL algorithms employed in our evaluation, referring to the original papers for exhaustive details (Wang et al., 2021; Yu et al., 2022; Gu et al., 2021; Bettini et al., 2024).

**QPLEX** (Wang et al., 2021). QPLEX is a value-based method designed for cooperative MARL. It builds on the QMIX framework by introducing a dueling network architecture. Each agent maintains its own local $Q$-utility, while a mixing network combines these into a joint action-value function. This decomposition allows decentralized execution while maintaining centralized training. Similar to DQN Mnih et al. (2013) in the single-agent case, QPLEX is restricted to discrete action spaces, making it applicable to topology optimization tasks.

**MAPPO** (Yu et al., 2022) and **LagrMAPPO** (Gu et al., 2021). MAPPO extends PPO (Schulman et al., 2017) to the multi-agent setting using a centralized critic and decentralized actors. Each agent learns its own policy, while the centralized critic leverages global information to centralize training. The clipped surrogate objective from PPO ensures stable updates, balancing policy improvement and regularization. MAPPO can handle both discrete (topology optimization) and continuous (redispatching and curtailment) actions, depending on the distribution chosen for the actor. LagrMAPPO augments MAPPO with a constraint-handling mechanism: in addition to training the policy, it learns Lagrangian multipliers associated with each constraint (as in Section 2). Policy updates then take gradient ascent steps in $\pi$ and descent steps in $\lambda$, trading off constraint satisfaction and task performance. This ensures penalties grow when constraints are violated and decay when constraints are respected.

**MASAC** (Bettini et al., 2024). MASAC adapts SAC (Haarnoja et al., 2018) to the multi-agent setting, combining centralized critics with decentralized actors. As in the single-agent SAC, MASAC jointly optimizes for expected return and policy entropy, encouraging exploration and robustness. Each agent learns a stochastic policy, while the centralized critic leverages information across agents to reduce variance and improve stability. MASAC supports well continuous action spaces and is therefore particularly suitable for our redispatching and curtailment tasks.

## B  STATE-SPACE

Tables 7 and 8 describe the remaining task-specific features composing agents' observations in the discrete and continuous case, respectively.

Table 7: List of additional features composing the state of a power grid for the discrete case. For brevity, `n_` indicates "number of", `gen,  sub` stands for "generators" and "substations", respectively, and `dim_topo` is the size of the vector containing the current topology of the grid.

| Name(s) | Type | Dim. | Description |
|---|---|---|---|
| `t` | int | 1 | Current simulation step |
| `gen_theta` | float | `n_gen` | Gens real power and voltage angle |
| `load_`$\theta$ | float | `n_load` | Loads active load and voltage angle |
| `topo_vect` | int | `dim_topo` | Topological vector of the grid; the bus to which each object is connected |
| `time_before_cooldown_line` | int | `n_line` | Line cooldown timer |
| `time_before_cooldown_sub` | int | `n_sub` | Cooldown timer for substations |
| `{time, duration}_next_maintenance` | int | `n_line` | Remaining time and duration of the next maintenance |

Table 8: List of additional features composing the state of a power grid for the continuous case. For brevity, `n_` indicates "number of", `gen, stor` stands for "generators" and "storage units", respectively.

| Name(s) | Type | Dim. | Description |
|---|---|---|---|
| `month` | int | 1 | Month of the year |
| `day_of_week` | int | 1 | Day of the week |
| `hour_of_day, minute_of_hour` | int | 1 | The time it is |
| `p_or` | float | `n_line` | Active power of each line |
| `storage_charge` | float | `n_stor` | Storage units charge |
| `storage_power` | float | `n_stor` | Storage units power |
| `curtailment` | float | `n_gen` | Curtailed power for each generator |
| `curtailment_limit` | float | `n_gen` | Limit imposed on each renewable generator |
| `gen_p_before_curtail` | float | `n_gen` | Production there would have been without curtailment |
| `target_dispatch` | float | `n_gen` | Targeted redispatching |
| `actual_dispatch` | float | `n_gen` | Implemented redispatching |

## C  AGENT CONFIGURATIONS

Table 9 reports the agent grid partitions for the *bus14* and *bus36* topology optimization (discrete) tasks. For these smaller grids, we focus exclusively on the discrete setting, which is substantially more challenging and already causes common MARL algorithms to struggle, even in the simplest *bus14* setup (see Section 4). By contrast, redispatching and curtailment (continuous) setups already achieve promising performance in the larger and more complex *bus118* scenario, making the smaller cases not challenging enough to investigate in the continuous setting.

Table 9: Agent-to-substation assignments, number of controlled components, observation and action dimensions for the local observation setup of *bus14, bus36* (T stands for the topology case)

| Grid | Agent | Controlled Substations | Lines | Gens. | Loads | \|Obs (T)\| | \|Actions (T)\| |
|---|---|---|---|---|---|---|---|
| bus14 | 0 | [0, 1, 2, 4] | 8 | 3 | 3 | 71 | 61 |
| | 1 | [3, 6, 7, 8] | 9 | 1 | 2 | 49 | 55 |
| | 2 | [5, 9, 10, 11, 12, 13] | 9 | 2 | 6 | 83 | 89 |
| bus36 | 0 | [0, 1, 2, 3, 4] | 9 | 1 | 6 | 77 | 77 |
| | 1 | [6, 7, 8, 9, 16] | 18 | 7 | 5 | 150 | 65642 |
| | 2 | [5, 10, 11, 12, 13, 14, 15, 32, 35] | 13 | 3 | 12 | 139 | 127 |
| | 3 | [17–31, 33, 34] | 32 | 11 | 14 | 377 | 1119 |

## D  REWARD

In this section, we formally define the reward components for the discrete topological tasks. We recall the joint reward the agents get at each step is $R = \alpha R_{\text{survive}} + \beta R_{\text{overload}} + \eta R_{\text{cost}}$. While $R_{\text{survive}}$ is a cumulative positive constant, the overload and cost rewards are defined as:

*(i) Overload*: Penalizes line overloads and disconnections, and rewards available line capacity based on the difference between line flows and capacity limits. In unconstrained settings, disconnected lines incur a fixed penalty. This is more formally defined as:

$$R_{\text{overload}} = \sum_{\ell \in \mathcal{L}} \left[ \max \left( 0, \frac{P_{F,\ell} - P_{F,\ell}^{\max}}{P_{F,\ell}^{\max} + \epsilon} \right) - \mathbb{1}(\ell \text{ is disc.}) \right], \tag{1}$$

where $P_{F,\ell}$ is the power flow on line $\ell$, $P_{F,\ell}^{\max}$ is its capacity limit, $\epsilon$ is a small constant to avoid divisions by 0, and the indicator function returns 1 if the line is disconnected. This term is then normalized to lie within $[-1, 1]$.

*(ii) Cost*: This component accounts for redispatching, curtailment, and storage usage, all of which induce operational costs. It is defined as:

$$R_{\text{cost}} = -\left[(P_G - P_D) + |c_{\text{redisp}}| + |P_{\text{storage}}|\right] c_{\text{marginal}},$$

where $P_G$ and $P_D$ denote the total power generated and total demand consumed at each step, respectively, with their difference representing transmission losses, $c_{\text{redisp}}$ corresponds to the redispatched power (i.e., the absolute deviation from scheduled generator setpoints), and $P_{\text{storage}}$ represents the power exchanged with storage units. All cost components are scaled by the marginal generation cost $c_{\text{marginal}}$, defined as the cost per MWh of the most expensive generator currently producing power. This value is also normalized to lie in the range $[-1, 0]$.

## E  HYPERPARAMETERS

Table 10 lists the hyperparameters considered during our initial grid search and the best-performing parameters used to run the experiments in Section 4.

Table 10: Details of the grid search used to find the best-performing hyperparameters for each algorithm in the topology optimization (discrete) and redispatching and curtailment (continuous) cases.

| Algorithm | Parameter | Grid search | Chosen value |
|---|---|---|---|
| **Shared** | *N° parallel envs* | 10, 20, 50 | 10 |
| | *Max gradient norm* | 10, 20, 50 | 10 |
| | *Discount $\gamma$* | 0.9, 0.95, 0.99 | 0.99 |
| | $\rho_{max}$ | 0.9, 0.95 | 0.9 |
| **Top. opt. reward** | $\alpha$ | 0.1, 0.5, 1.0 | 1.0 |
| | $\beta$ | 0.1, 0.5, 1.0 | 1.0 |
| | $\eta$ | 0.1, 0.5, 1.0 | 1.0 |
| **QPLEX** | *Train frequency* | 10, 50, 100 | 100 |
| | *Target network update* | 250, 500, 2500 | 2500 |
| | *Buffer size* | 500000, 1000000 | 1000000 |
| | *Batch size* | 128, 256 | 128 |
| | *Learning rate* | 0.003, 0.0003, 0.00003 | 0.00003 |
| | $\epsilon$-*decay fraction* | 0.1, 0.25 0.5 | 0.5 |
| **MAPPO** | *N° steps (total)* | 10000, 20000, 40000 | 20000 |
| (discrete case) | *N° minibatches* | 1, 4, 8 | 4 |
| | *N° update epochs* | 20, 40, 80 | 80 |
| | *Actor learning rate* | 3e-3, 3e-4, 3e-5 | 3e-5 |
| | *Critic learning rate* | 3e-3, 3e-4, 3e-5 | 3e-5 |
| | $\epsilon$-*clip* | 0.1, 0.2, 0.3 | 0.2 |
| **MAPPO** | *Batch size* | 3000, 6000, 9000 | 9000 |
| (continuous case) | *N° update epochs* | 5, 15, 30 | 30 |
| | *Actor learning rate* | 3e-4, 3e-5, 3e-6 | 3e-5 |
| | *Critic learning rate* | 3e-4, 3e-5, 3e-6 | 3e-5 |
| **LagrMAPPO** | $\lambda$ | 0, 50 | 0 (L), 50 (O) |
| | $\lambda$ *init* | 0.0, 1.0 | 0.0 |
| | $\lambda$ *learning rate* | 0.01, 0.025, 0.05 | 0.05 |
| **MASAC** | *Batch size* | 3000, 6000, 9000 | 9000 |
| | *Minibatch size* | 128, 256 | 256 |
| | *N° optimizer steps* | 1000, 2000 | 1000 |
| | *Learning rate* | 3e-4, 3e-5, 3e-6 | 3e-4 |

## F    ADDITIONAL PLOTS FOR SECTION 4

To complement the main results in the topology optimization (discrete) case, we evaluate the best-performing baseline, MAPPO, on the more complex *bus118* system. Unlike in the smaller *bus14* grid, where MAPPO manages to sustain operation for a substantial fraction of the episode horizon, performance on *bus118* is unsatisfactory. Figure 7 summarizes the outcomes in terms of average survival at training time and analyzes the margin and topology scores for the trained policies. Survival rates are close to zero, indicating that MAPPO fails to maintain stable operation for more than a few steps. This is reflected in the margin metric, which remains consistently low and shows that agents are unable to preserve sufficient transmission capacity to handle contingencies. Similarly, the topology score indicates that agents rarely exploit meaningful structural reconfigurations; deviations from the initial configuration are minimal and do not translate into improved stability.

Overall, these results highlight the dramatic increase in difficulty when scaling from *bus14* to *bus118*. Even our strongest baseline fails to discover effective strategies for coordinated topology optimization at this scale, reinforcing the conclusion that MARL-based grid control requires new algorithmic advances beyond current MARL literature.

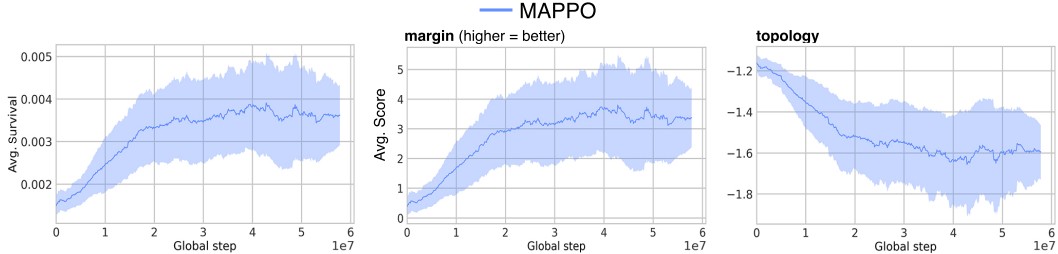

Figure 7: Results of the best performing baseline, MAPPO, in the topology optimization (discrete) bus118 task. (Left) Average survival during training for the discrete case on the bus14 task. (Center) Avg. margin score for the trained policy. (Right) Avg. topology score for the trained policy.

Moreover, Figure 8 presents the same operational metrics analysis as Figure 5, but for the constrained baseline. LagrMAPPO with load shedding and islanding constraints (L) achieves higher performance than the transmission line overload constrained version (O), despite operating under a stricter threshold. Notably, these policies tend to converge on a single topological modification that increases available margins, allowing the grid to remain operational for roughly 20% of the episode horizon.

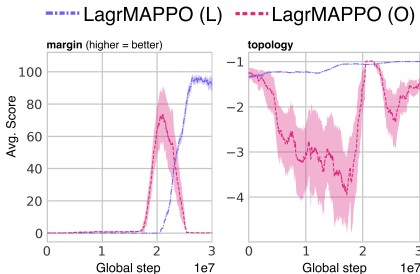

Figure 8: Average score for line margins and topological changes for the constrained algorithm of Figure 4.

