# OpenReview forum: "MARL2Grid-TR: A Multi-Agent RL Benchmark in Power Grid Operations"
_ICLR.cc/2026/Conference — ICLR 2026 Poster_

### Official Review · Reviewer_2m4V · 2025-10-30

**Soundness:** 3
**Presentation:** 4
**Contribution:** 4
**Rating:** 8
**Confidence:** 2

**Summary:**

The paper proposes a MARL simulation framework for power grid operations. By using realistic scenarios, it proposes a tool for evaluating
decentralized control of substations and generators. It supports various configurations, including partial information, delayed rewards, and physical constraints.

**Strengths:**

- The first standardized MARL formulation for topology optimization and redispatching control, including safety constraints
- Integration with PETTINGZOO for MARL compatibility and reproducibility.
- Baseline evaluation of state-of-the-art MARL algorithms
- Simulation environments based on real TSO data and topologies.
- Fills an important gap - since previous power grid RL environments were single-agent. The combination of high-fidelity grid simulation and multi-agent formalism is novel and impactful.

**Weaknesses:**

- The title is misleading - the paper offers a simulation environment, not a rigorous account of how benchmarking should be performed (i.e., KPIs etc).
- Evaluation focuses mainly on classic CTDE baselines with no ablation studies on the effect of observability regimes (centralized vs. local).
- No explicit evaluation metrics on economic cost, safety violation frequency, or stability margins over time.
- Minor writing issues (redundancy, figure clarity, inconsistent notation in equations).
- Evaluation focuses on different SOTA algorithms, but since the focus is on the simulation, I would expect an account of its limitations.

**Questions:**

- What are the limitations of the simulation in terms of the algorithms and the settings that it can support? There is an extensive discussion on why the current algorithms fail to address the complexities, but no account of the limitations of the simulation itself.

- In the context of constraint handling, did you analyze the patterns of violations in different settings, and did you experiment with agent-local constraints?

- While I accept the CTDE paradigm here, can you speculate on the result of applying a fully decentralized approach?

---

> ### Author Response · Authors · 2025-11-20
>
> We appreciate the useful comments and the positive evaluation of our work. In the following, we clarify the weaknesses (W) and questions (Q) raised by the reviewer. Notably, we have incorporated this discussion in the revised manuscript (highlighting changes with blue text).
>
> **W1: “...title is misleading…”:**
>
> We agree that the title may suggest a broader focus on benchmarking methodology (e.g., standardized KPIs, evaluation protocols) rather than on the simulation environment and task suite we contribute. Our intent was to emphasize that MARL2Grid provides benchmark tasks, standardized scenarios, and reference baselines, rather than a prescriptive benchmarking framework. We have now changed the title to “MARL2Grid-TR: A Multi-Agent RL Benchmark in Power Grid Operations.”
>
> **W2: “...evaluation focuses on classic CTDE baselines…”:**
>
> We appreciate the reviewer’s suggestion. In the revised version, we now include a single-agent PPO and its Lagrangian version with full observability. This baseline removes all decentralization effects and provides the strongest possible centralized reference. It observes the grid state, controls all substations and generators directly, and is trained under the same reward functions and transition dynamics as the MARL agents. As shown in the experiments, this centralized controller consistently underperforms the multi-agent MAPPO baseline for the topological task, even on the bus14 grid, and performs significantly worse in the constrained settings. On the contrary, for the redispatching and curtailment task, the single-agent PPO performs better than the MAPPO agent while being significantly less sample efficient (we assess this result as being due to the “reasonable” problem size of the redispatching task compared to the topological one).
>
> We also note that our benchmark is fully configurable in its observability settings: users can switch between fully centralized, partially shared, or strictly local observations through simple configuration changes. While our experiments focus on the regime our TSO collaborator identified as most realistic, CTDE training with decentralized execution, researchers are free to explore any observability setup their methods support.
>
> **W3: “...no explicit evaluation metrics…”:**
>
> We appreciate the reviewer’s observation. While incorporating additional evaluation metrics would certainly be valuable, doing so would introduce substantial additional complexity and broaden the scope beyond what we intended for this initial benchmark release. Our primary goal was to establish MARL tasks and highlight the fundamental algorithmic challenges. That said, we fully agree on the importance of richer evaluation metrics and, in fact, already discuss these directions in the original manuscript (Section 5). We view them as natural extensions for future versions of the benchmark and for researchers building on our work. As previously mentioned, our codebase allows the user to add these metrics in order to optimize/design algorithms for different objectives than the one originally proposed for benchmarking.
>
> **W4: “...minor writing issues…”:**
>
> We thank the reviewer for pointing this out. We will carefully proofread the entire manuscript to remove redundancy, improve figure clarity, and ensure consistent notation across equations and tables. If the reviewer has specific examples in mind, we would be grateful for the additional detail to ensure that we address all concerns precisely; otherwise, we will proceed with a thorough editing pass in the revision.
>
> **W5 + Q1: “...limitations of the simulation…”:**
>
> We agree that the simulation’s limitations should be made explicit. We briefly summarize this in the following:
>
> - Physical modeling: our benchmark builds on Grid2Op’s AC power-flow solver, which captures key operational constraints but does not include fast electromagnetic transients, detailed inverter dynamics, or full protection-system behavior (although it is possible to change the underlying power simulation framework to increase realism).
> - Action abstractions: topology actions model substation switching but omit certain practical constraints (e.g., breaker wear, personnel availability). Redispatch and storage models remain simplified to match supervisory control timescales.
> - Scenario realism: ChroniX2Grid time series and contingency models are realistic but cannot fully replicate all weather-driven or protection-system events in large grids.
> - Computational scalability: Larger grids can be loaded via configuration files, but MARL training on very large systems (>1000 buses) may become computationally intensive, though DC approximations can help mitigate this.
>
> We agree that making these limitations explicit helps clarify what our benchmark faithfully represents and where its scope ends, complementing our analysis of algorithmic failures and supporting its role as a realistic testbed for multi-agent grid control. We have now done so in the manuscript.

---

> > ### Author Response · Authors · 2025-11-20
> >
> > **Q2: “... the patterns of violations…”:**
> >
> > While our TSO collaborators emphasized that real operators aim to minimize the system-wide impact of outages (i.e., by distinguishing between full and partial blackouts), the underlying Grid2Op’s current modeling treats any load or generator loss as a game-over event, which also motivates our choice of joint constraints and explains why localized constraints were not the focus of this initial benchmark.
> >
> > Regarding violation patterns, we observed that failures are typically dominated by global overload cascades rather than localized issues, reinforcing the suitability of the joint-constraint formulation. For these reasons, we did not evaluate agent-local constraints in this initial benchmark. We view exploring localized constraints as a possible avenue for future work, whereas the joint version matches real-world practice and thus forms the foundation of our current experiments.
> >
> > **Q3: “...can you speculate on the results of applying a fully decentralized…”:**
> >
> > We thank the reviewer for the interesting question. For the topological task, we expect a fully decentralized IPPO to achieve a sharp drop in performance compared to MAPPO due to the increased complexity in terms of exploration, non-stationarity and credit assignment.
> >
> > For the continuous setup, we can go beyond speculation: in separate preliminary experiments on the bus118 task (that goes beyond the scope of our paper), we tested two alternative approaches: (i) IPPO, and (ii) a sequential training scheme where each local agent is trained with PPO as a single agent, after which the redispatching agent is trained while keeping the local policies fixed. This second approach largely avoids the credit-assignment issues caused by partial observability. Despite achieving promising results, IPPO underperforms MAPPO, which further motivates our choice of a CTDE training framework. Interestingly, the fully decentralized single-agent RL approach outperforms MAPPO and gets comparable performance to the fully observable PPO added in the paper revision. This experiment indicates that credit assignment is one of the main factors hindering learning in redispatching tasks, while coordination in the continuous 118-bus is not particularly problematic under our setup.
> > ___
> > Once again, we thank the reviewer for the insightful comments. We incorporated the above discussion in the revised paper and remain available for any further clarification.

---

### Official Review · Reviewer_eRUd · 2025-10-31

**Soundness:** 3
**Presentation:** 3
**Contribution:** 3
**Rating:** 4
**Confidence:** 4

**Summary:**

This paper identifies the limitation of prior work in power grid operations, which has largely focused on single-agent settings, and proposes MARL2GRID, the first benchmark for multi-agent reinforcement learning (MARL) designed to reflect the decentralized nature of real-world grid control. Built upon the high-fidelity Grid2Op simulator, the benchmark introduces realistic tasks, including discrete topology optimization and continuous redispatching, along with safety-critical constraints. Through experiments, the authors demonstrate that existing MARL algorithms (e.g., MAPPO, QPLEX) struggle significantly with the scalability, coordination, and safety requirements of these tasks, particularly in the combinatorially complex topology optimization setting, thereby highlighting key challenges for future research.

**Strengths:**

1, The paper correctly identifies a critical gap between existing single-agent RL research (e.g., L2RPN) and the decentralized reality of power grid control, making the proposal of a MARL-specific benchmark both timely and significant. The design, developed in collaboration with Transmission System Operators (TSOs), adds a layer of realism and credibility to the proposed problem setting.

2, The benchmark is well-structured, offering a suite of tasks across various grid scales (bus14, bus36, bus118), two distinct and crucial control paradigms (discrete topology and continuous redispatch), configurable observability regimes, and explicit safety constraints such as load shedding and line overloads (Sections 3, 3.2). This provides a rich and challenging testbed for future MARL research.

3, The empirical results effectively highlight the brittleness of current MARL algorithms when faced with a complex, realistic task. For instance, while MAPPO achieves a 79% survival rate on the bus14 topology task (Table 5), its performance collapses to near-zero on the more complex bus118 grid (Figure 7), starkly illustrating the benchmark's difficulty and motivating the need for novel algorithmic solutions.

**Weaknesses:**

1, The paper's central premise is that the decentralized nature of power grids necessitates a MARL approach (Lines 014-016). However, this claim is not substantiated with experimental evidence. A critical baseline is fatally absent: a strong, centralized single-agent RL controller (e.g., single-agent PPO) that has access to the full state and action space. Without this comparison, it is impossible to determine whether the proposed multi-agent decomposition (1) offers any performance benefit, (2) is truly necessary for scalability, or (3) merely introduces unnecessary complexity that hinders performance. This is a major flaw in validating the paper's core motivation.

2, The paper successfully demonstrates that current algorithms fail, particularly on large-scale topology optimization (Figure 7). However, it stops at reporting the phenomenon (e.g., "MAPPO failed") without providing a deep diagnostic analysis of why they fail. The root cause—be it challenges in credit assignment under partial observability, inefficient exploration in a combinatorial action space, or poor coordination—remains unexplored. This lack of insight limits the benchmark's utility in guiding future

**Questions:**

1, The role and efficacy of the "idle heuristic" are unclear. Was this heuristic only applied in the bus118 continuous task, or was it also used for the bus14 topology task where MAPPO performed well? To isolate and validate its contribution, could you provide an ablation study (e.g., MAPPO with vs. without the heuristic on the bus14 task) to quantify its actual impact on performance?

2, To experimentally validate the core motivation for a MARL approach, could you please provide a performance comparison against a centralized single-agent RL baseline (e.g., single-agent PPO) on the bus14 and bus118 topology optimization tasks? This result is crucial for justifying the choice of the MARL paradigm over a simpler, centralized one.

3, Performance drops dramatically from bus14 to bus118. To better understand the scalability limits of current MARL methods, could you provide results for the intermediate-sized bus36 grid in the topology optimization task? This would help clarify whether the performance degradation is gradual or a sharp cliff-edge effect at a certain complexity threshold.

4, The results show that algorithms fail on large-scale tasks (Figure 7). Can you provide a more in-depth diagnostic analysis of these failures? For instance, is the failure primarily due to a lack of coordination, inefficient exploration, or credit assignment challenges under partial observability? A statistical breakdown of which safety constraints are most frequently violated would provide valuable guidance for future research.

---

> ### Author Response · Authors · 2025-11-20
>
> We thank the reviewer for their thoughtful feedback and would like to address the weaknesses (W) and questions (Q) raised and highlight our recent efforts toward further improving our work. We have also incorporated the following discussion in the revised paper (highlighting changes with blue text).
>
> **W1 and Q2: “...provide a strong, centralized single-agent RL controller (e.g., single-agent PPO) comparison…”:**
>
> We agree that evaluating a fully-observable single-agent PPO would further validate one of the paper’s core motivations. However, we would like to emphasize that scalability is not our only motivation. In fact, the dimensions of the redispatching-curtailment task remain reasonable on the 118-node grid. The second important motivation is the desire to be realistic and respect the decentralized aspect of power grid operation. For the topological task, we expect decentralization to help us manage the large dimensions of the problem, but for the continuous task, we are in fact making its resolution more complex.
>
> We conducted the requested PPO (and LagrPPO) experiments. For the topological task, we found that PPO performs worse than the MAPPO baseline, even in the relatively small bus14 case. This result holds for both the unconstrained and constrained formulations: while LagrMAPPO substantially reduces safety violations (compared to LagrPPO), the single-agent PPO struggles to discover effective topology adjustments and accumulates more constraint violations.
> To further validate this, we extended our policy analysis (Figure 5) to the PPO policy and observed exactly the expected pattern: the PPO agent performs fewer topological reconfigurations and maintains significantly lower line-loading margins compared to MAPPO. These findings reinforce that decentralization is a valuable addition to learning-based topological power grid control.
> For the redispatching and curtailment bus118 task, PPO performs slightly better than decentralized agents at convergence (due to the large scale and the need to deal with the credit assignment issue and cooperative learning). Nonetheless, PPO also required 200% more steps to reach convergence and such performance. We therefore consider the MAPPO results to be encouraging for future work.
>
> We believe these additional results provide sufficient evidence for justifying the choice of the MARL paradigm. In this direction, we also want to remark how the multi-agent zonal structure of our benchmark has been informed by our collaboration with a TSO and thus reflect their interest and operations.
>
> **W2 and Q4: “...an analysis of why algorithms fail…”:**
>
> The reviewer raised a valid point, and we agree that our original presentation did not sufficiently connect the observed failures to the underlying problem structure. While these challenges were discussed abstractly in Section 5, they were not explicitly connected to our experimental evaluation. Below we summarize the main reasons why current methods fail, based on our empirical observations and the physics of power grids.
> 1. Exploration “collapses” in the exponentially large topology action space: a single substation can have tens of thousands of configurations, making beneficial multi-step reconfigurations extremely rare. Moreover, exploration is really difficult in practice as random actions are generally bad and lead to the end of an episode in a few time steps (e.g., a random policy has an average episode length of 10 steps). This is also discussed in [D]. Therefore, designing more structured exploration approaches is one of the keys to successful training for these kinds of physical tasks.
> 2. Coordination breaks down across electrically coupled zones: actions that improve margins locally often overload distant lines. This is a non-relevant phenomenon in bus14 but dominant in larger grids.
> 3. Credit assignment becomes unreliable due to delayed global penalties, preventing agents from linking distant or delayed outcomes to their own decisions.
> 4. Long-horizon irreversible dynamics often send the system into unrecoverable states. This is often caused by early random switches triggering cooldown timers, islanding risks, and the overload-to-disconnection logic.
>
> Together, these are the main factors that prevent policies from increasing margins or discovering meaningful topology changes in larger grids, explaining the sharp performance drop. Finally, recent works referenced in our manuscript (e.g., RL2Grid) already show that even a fully observable single-agent PPO controller fails on bus118, confirming that the difficulties 1-4 stem from the intrinsic structure of realistic topology control rather than from decentralization. We now highlight this explicitly in Section 4.1 of the revised paper, which links our observed failure modes to the research directions outlined in Section 5.

---

> > ### Author Response · Authors · 2025-11-20
> >
> > **Q1: “...efficacy of the “idle heuristic” are unclear…”:**
> >
> > We thank the reviewer for this question. The point is valid, and we agree that the role of the idle heuristic should be made more evident. The ablation the reviewer requests was already included in the paper, but we recognize that it was not highlighted clearly enough. In the revision, we make this explicit in Section 4 and clarify that MAPPO is evaluated both with and without the idle heuristic on the bus14 topology task, where we directly compare the two variants. We also added a discussion to make the heuristic’s impact on the results more clear.
> >
> > **Q3: “...performance drops dramatically from bus14 to bus118…”:**
> >
> > ​​We appreciate the reviewer’s suggestion. In principle, bus36 could serve as an intermediate complexity point. However, in practice, bus36 is not an intermediate case between bus14 and bus118 in the topology optimization setting. If we consider a centralized controller, bus14 has 209 total unique actions, bus36 has ~67000, and bus118 has ~72000 actions. Specifically, bus36 contains the same extremely large and complex substation structure that drives the action-space explosion in bus118 (see Tables 3 and 9).
> > ___
> > Overall, we thank you for your comments and insights, which have been helpful in further improving the paper.
> > ___
> > [D] D. Yoon et al., “Winning the L2RPN challenge: power grid management via semi-markov afterstate actor-critic,” ICLR, 2021.

---

> > > ### Comment · Reviewer_eRUd · 2025-11-28
> > >
> > > I thank the authors for their detailed response. The addition of the single-agent PPO baseline effectively validates the necessity of the MARL approach, which was my primary concern. I also appreciate the deeper analysis of the failure modes in the revised manuscript. As my major concerns have been fully resolved, I am raising my rating to 6.

---

### Official Review · Reviewer_q5yx · 2025-11-01

**Soundness:** 3
**Presentation:** 3
**Contribution:** 2
**Rating:** 4
**Confidence:** 4

**Summary:**

Modern power grids face increasing complexity due to renewable integration, requiring fast, decentralized control beyond traditional optimization. Prior RL benchmarks (like L2RPN, RL2GRID) model the grid as a single-agent problem, ignoring real-world decentralization. The paper introduces MARL2GRID, the first benchmark for multi-agent RL (MARL) in power grid operations, developed with transmission system operators (TSOs).

**Strengths:**

-	Power grid control is challenging and important especially with more renewable integration. RL holds a big promiss, and this paper addresses a key gap for benchmarking decentralized MARL for realistic grid operations.
-	This paper is built on industrial-grade Grid2Op simulations with realistic dynamics, long horizons, and stochastic disturbances.
-	This paper considers safety constraint, which is important for grid operations.

**Weaknesses:**

-	More specifics about the intended power system use case. From my understanding, power system typically have a 3-layered control architecture, with primary control (droop) occurring at the fastest time scale, and then secondary control (AGC), followed by tertiary control. What time-scale is the benchmark and how it would fit into the existing control architecture of power systems?
-	Scale of the benchmark. For applications like power systems, scale matters a lot and I would like to see a much larger scale (in the thousands/10k’s sized) power system benchmarks.
-	The proposed benchmark also lacks diverse tasks to be named (MARL2GRID). As I mentioned, there are many control problems happening at the grid at different levels, including lower/fast level frequency/voltage control control at generators/inverters, mid level AGC as well as higher level day-ahead planning. For the benchmark to be named MARL2GRID, ideally it should consider more than 1 tasks that are representative of the complex grid operations.

**Questions:**

I'd like to see a thorough discussion on how RL can be deployed in grid operations. Power grid is a very conservative industry, and it is more or less reluctant to emerging technologies, especially if existing technologies do just fine. I'd like to see some more discussions on what RL can achieve that traditional methods cannot.

---

> ### Author Response · Authors · 2025-11-20
>
> We appreciate your insights and would like to highlight the revisions we made to the paper to address the weaknesses (W) and questions (Q) raised.
>
> Before we dive in, we want to emphasize why having a MARL benchmark for power grid operations matters. There is currently no scalable solver for large-scale topology optimization, leaving TSOs without an algorithmic alternative for one of their most critical (and cost effective) control levers. Our goal is to leverage the enthusiasm of early-adopter TSOs that collaborated on this work, to demonstrate and advance the feasibility and safety of learning-based approaches, and to do so in a way that builds the evidence and confidence needed for more conservative operators to eventually adopt these methods as well.
>
> **W1: “...specifics about the intended power system use case…”:**
>
> We thank the reviewer for raising this important point, which could be a source of misunderstandings. In the underlying Grid2Op framework on which we built our benchmark, primary frequency control is done by the backend (i.e., PandaPower or Lightsim2Grid, in our case), secondary control is done by a redispatching routine, and the agent operates at the tertiary control level for both discrete and continuous actions. Broadly speaking, the agent currently models the operator and, in general, primary and secondary frequency control are automatic. Notably, tertiary control is also the layer where RL has shown promise in prior single-agent L2RPN/RL2Grid works referred to in the manuscript.
>
> Thus, our benchmark reflects the supervisory/tertiary control layers that our TSO collaborator identified as the most challenging and relevant for learning-based decision support.
>
> **W2: “...scale of the benchmark…”:**
>
> We agree that scale is critical for practical grid applications, but the benchmark’s goal is also to reflect the current reality of MARL research in the power grid tasks we consider. We’d like to emphasize three important points:
>
> - To our knowledge, most MARL studies such as [A, B] try to provide algorithmic solutions for up to 5, 14 buses (i.e., without providing a benchmark and leaving 36 bus scenarios as future work), while our benchmark extends our considerations to a 118-bus network. Even in single-agent RL, most works (e.g., the L2RPN, RL2Grid mentioned in the manuscript) use environments with 118 buses or less [C], and they do so without decentralization or partial observability.
> - Our results show that moving from 14 to 118 buses already exposes fundamental unsolved challenges (e.g., exponential topology action spaces, long-horizon safety dynamics, credit assignment and cross-zone coordination under partial observability) indicating that these difficulties arise well before thousand-bus systems.
> - Although we focus on up to 118 buses, our benchmark is built on Grid2Op and can directly support larger grids via configuration files, placing no structural limit on scale. We also note the non-negligible computational demands of running a power flow at each step of the environment. As discussed in the paper, we thus believe more algorithmic advances are needed before scaling up to larger grids.
>
> Philosophically, we believe the purpose of our benchmark is to meet methods where they currently stand and push them one level further before advancing to the next. In this sense, the 118-bus setting is already a meaningful and demanding step up for MARL with tens of thousands of actions on the topological discrete case (which is already a significant challenge for RL). This setup is large enough to capture realistic operator challenges, yet still tractable for systematic experimentation. Scaling to thousand-bus systems remains an important future direction, but our findings make clear that substantial algorithmic advances are needed even before reaching that scale.
>
> **W3: “...lacks diverse tasks to be named (MARL2GRID)...”:**
>
> We agree that the name MARL2Grid may suggest broader coverage than the tasks currently included. To make the scope explicit, we are renaming the benchmark to MARL2Grid-TR, highlighting its focus on *T*opology optimization and *R*edispatching/curtailment & storage. We are also happy to consider alternative names that the reviewer feels would better communicate the benchmark’s intended focus.

---

> > ### Author Response · Authors · 2025-11-20
> >
> > **Q1: “...how RL can be deployed in grid operations…”:**
> >
> > We appreciate the reviewer’s request for a clearer discussion of how RL could be deployed in power-grid operations, especially in a sector that is appropriately conservative and highly regulated. On top of our initial remark, we believe the related single-agent RL works RL2Grid and L2RPN challenges (organized by a TSO), along with our discussion with the TSO who has co-designed our benchmark, highlight the concrete interest in these approaches. Another interesting discussion is provided in the “Paris Region AI Challenge for Energy Transition” article which shows that both the Paris Region and RTE (France’s TSO) are aiming at deploying these kinds of methods to help operators make better decisions in real time.
> >
> > It is also important to remark that traditional optimization and rule-based methods are increasingly strained by modern operating conditions: high VRE penetration creates fast, uncertain flow patterns; flexible loads and storage add new coupled dynamics. Thus, TSOs are looking for new methods and RL is gaining popularity due to its ability to consider temporal dependencies. As a result, RL is viewed by TSOs not as a replacement for established tools, but as a potential complement that can amortize long-horizon nonlinear decision making, discover non-intuitive corrective actions, and provide data-driven decision support under uncertainty.
> > ___
> > We have now incorporated the above discussion in the revised manuscript. Overall, we thank the reviewer for their comments, as they have helped us further improve the clarity and presentation of our work.
> > ___
> >
> > [A] E. van der Sar et al., “Multi-Agent Reinforcement Learning for Power Grid Topology Optimization,” arXiv, 2023.
> >
> > [B] B. de Mol et al., “Centrally Coordinated Multi-Agent Reinforcement Learning for Power Grid Topology Control,” E-Energy, 2025.
> >
> > [C] E. van der Sar et al., “Optimizing Power Grid Topologies with Reinforcement Learning: A Survey of Methods and Challenges,”arXiv, 2025.

---

### Author Response · Authors · 2025-11-25

Dear Reviewers,

We are reaching out to help foster discussion during the remaining days of the rebuttal period and to highlight the revisions we made in direct response to your thoughtful feedback. Your comments have already led to substantial improvements, and we want to ensure that all concerns have been fully addressed.

If there are any points you would like us to expand on, or if further clarification would help strengthen the final version, we would be very happy to continue the conversation. Your engagement has been extremely valuable, and we genuinely appreciate the time and constructive input you have provided.

---

### Author Response · Authors · 2025-12-01

We would like to thank the AC who handled our submission so far, the newly assigned AC, and all reviewers for their time, effort, and thoughtful engagement with our work. Following the conference’s recent guidance, we provide here a concise overview of the revisions made in direct response to reviewers' feedback.

During the rebuttal period, we revised the manuscript with new experiments, expanded analyses, and improved overall clarity. In particular, we:

- *Clarified the intended power-system control layer* of our work, renaming the benchmark to **MARL2Grid-TR**, making its scope more explicit: **T**opology optimization and **R**edispatching/curtailment & storage.
- *Expanded the scalability discussion*, explaining why the 118-bus system already goes beyond MARL literature and reveals core MARL challenges, how the benchmark compares to existing work, and how it can scale as methods mature.
- *Added a centralized single-agent PPO baseline (and its constrained version)* with full observability and full action authority. These baselines underperform MAPPO on topology tasks and are less sample-efficient for redispatching, reinforcing the value of the multi-agent formulation. This strengthened our work by further analyzing the role of decentralization and credit assignment.
- *Introduced a dedicated section analyzing failure modes*, such as exploration collapse, cross-zone coordination, credit assignment, and long-horizon irreversibility. We linked these insights to the research directions that are extensively discussed in Section 5. We also made the idle-heuristic ablation explicit and expanded its analysis.
- *Added a discussion on simulation limitations*, covering modeling assumptions, action abstractions, scenario realism, and computational constraints.
- *Clarified the rationale for using joint constraints*, reflecting TSO practice, and summarized the violation patterns observed in experiments.
- *Improved writing quality*, notation consistency, and figure clarity throughout.

We also note that Reviewer eRUd explicitly acknowledged the quality and completeness of these revisions by increasing their evaluation. Although Reviewer q5yx did not respond before the policy change, we are confident that the revisions comprehensively address their concerns as well.

We are grateful for the AC’s and reviewers’ contributions and appreciate the opportunity to refine the paper based on such constructive feedback.

---

### Meta-Review · Area_Chair_FtYh · 2026-01-08

**Summary:**

Reviewers agreed that the paper introduces a timely and impactful benchmark for multi-agent reinforcement learning in power grid operations, addressing an important gap left by prior single-agent benchmarks. Strengths include the use of high-fidelity industrial-grade Grid2Op simulations, realistic safety constraints, and experiments that clearly demonstrate the limitations of current MARL methods under realistic conditions. Initial concerns focused on clarity of scope and control layer, scale and task coverage, justification of decentralization, and the lack of diagnostic analysis explaining algorithmic failures.

**Reviewer Concerns:**

The rebuttal and revision substantially addressed the main concerns. The authors clarified the intended tertiary/supervisory control layer,  renamed the benchmark to MARL2Grid-TR, and adjusted the title to better reflect the paper scope. They added strong centralized single-agent PPO and constrained PPO baselines, expanded the analysis of decentralization trade-offs, introduced detailed failure-mode diagnostics, and clarified the role of the idle heuristic explicit. These additions significantly strengthened the paper’s validation and positioning.

Some limitations remain by design rather than omission. Empirical evaluation focuses on grids up to 118 buses and on topology optimization and redispatching, with larger-scale systems and additional control tasks left for future work. These choices are now clearly motivated and appear appropriate for the benchmark’s stated goals.

**Reviewer Scores:**

It is not easy to estimate how the reviewers would have been adjusted their scores even assuming a full discussion. However, one reviewer explicitly noted that their main concern had been addressed and increased their score from 4 to 6 during the rebuttal phase. The other reviewer who initially scored the paper at 4 would likely have kept or increased slightly their score, given that their primary concerns were carefully discussed. The reviewer with an initial score of 8 is unlikely to have substantially changed their evaluation. Overall, this would plausibly lead to an average score around 6, with discussion narrowing the initial disagreement and leaning towards a borderline-accept outcome.

---

### Decision · Program_Chairs · 2026-01-26

Accept (Poster)